# The Effect of Grip Width on Muscle Strength and Electromyographic Activity in Bench Press among Novice- and Resistance-Trained Men

**DOI:** 10.3390/ijerph18126444

**Published:** 2021-06-14

**Authors:** Atle Hole Saeterbakken, Nicolay Stien, Helene Pedersen, Tom Erik Jorung Solstad, Kristoffer Toldnes Cumming, Vidar Andersen

**Affiliations:** 1Department of Sport, Food and Natural Sciences, Faculty of Education, Arts and Sports, Western Norway University of Applied Sciences, N-6856 Sogndal, Norway; nicolay.stien@hvl.no (N.S.); helene.pedersen@hvl.no (H.P.); tom.erik.jorung.solstad@hvl.no (T.E.J.S.); vidar.andersen@hvl.no (V.A.); 2Faculty of Health and Welfare Sciences, Østfold University College, N-1757 Halden, Norway; kristoffer.t.cumming@hiof.no

**Keywords:** free weights, training status, resistance, EMG, strength

## Abstract

Background: This study compared the muscle activity and six repetition maximum (6-RM) loads in bench press with narrow, medium, and wide grip widths with sub-group comparisons of resistance-trained (RT) and novice-trained (NT) men. Methods: After two familiarization sessions, twenty-eight subjects lifted their 6-RM loads with the different grip widths with measurement of electromyographic activity. Results: Biceps brachii activity increased with increasing grip width, whereas wide grip displayed lower triceps brachii activation than medium and narrow. In the anterior deltoid, greater activity was observed using a medium compared to narrow grip. Similar muscle activities were observed between the grip widths for the other muscles. For the RT group, greater biceps brachii activity with increasing grip width was observed, but only greater activity was observed in the NT group between narrow and wide. Comparing wide and medium grip width, the RT group showed lower triceps activation using a wide grip, whereas the NT group showed lower anterior deltoid activation using a narrow compared to medium grip. Both groups demonstrated lower 6-RM loads using a narrow grip compared to the other grips. Conclusion: Grip widths affect both 6-RM loads and triceps brachii, biceps brachii, and anterior deltoid activity especially between wide and narrow grip widths.

## 1. Introduction

The bench press exercise is perhaps the most frequently used exercise to gain strength, hypertrophy, and power in the upper body among recreational, resistance-trained, and power lifters. Typically, the bench press exercise is performed lying on a bench while the barbell is lowered to the chest before being elevated until the elbows are extended [1]. To the authors’ best knowledge, few studies have examined the differences in bench press performance (e.g., muscle strength and muscle activation) between populations with different resistance training experience.

Several studies have examined the bench press exercise. For example, different chest-press exercises have been examined with regard to muscle strength, electromyographic activity (EMG) and kinematics [2,3], bench angles [4,5], biomechanics [6,7,8,9], unstable surfaces [10,11], successful and unsuccessful attempts [6], and different muscle actions (e.g., isometric, concentric only, and counter movement) [12,13]. The main focus has been the primary muscle groups (pectoralis major, deltoid anterior, and triceps brachii) in addition to the synergist and antagonist in the exercise [1,9,14,15].

The reported effects of grip width on muscle activity in barbell bench press have not been conclusive. For example, Saeterbakken et al. [4] demonstrated similar muscle activity in the pectoralis major, deltoids, and triceps brachii with the exception of lower biceps brachii activation in narrow and medium grips compared to wide grip among competitive bench press athletes. Corresponding biceps brachii activation was observed by Lehman et al. [12] comparing similar conditions, but they also demonstrated greater pectoralis major activity using a wide grip. Accordingly, Barnett et al. also reported a wide grip (i.e., 200% of the biacromial distance) to result in greater pectoralis major activity [16,17], but lower triceps brachii and anterior deltoid activation [17]. Most recently, Mausehund et al. [9] included 35 resistance-trained adults and demonstrated greater EMG activity in triceps brachii, anterior deltoid, and pectoralis major with a narrow grip than wider grip widths using 6–8RM loads. Different loads (i.e., 1-RM, 80% of 1-RM, 6–8RM), muscle contraction (i.e., dynamic, isometric), equipment (i.e., Smith machine, free weights), and resistance training experience (i.e., competitive, >2 years, >6 months) may explain the different findings. Furthermore, it is generally accepted that wider grip increases loads lifted due to shorter vertical displacement of the barbell [4,17]. While both different loads and muscle activation have been examined previously [18,19,20,21], the effects of training experience and muscle activation have been neglected in the literature.

Changes in both the muscles (e.g., cross-section area, fiber length, fiber pennation angle) and neural system (e.g., motor unit activation, firing frequency, and synchronization) can be examined to monitor improvements following resistance training, as well as to differentiate between novice and experienced lifters [22]. Considerable debate exists about the neurological changes that accompany strength improvement with investigations demonstrating both no change [23,24,25] and increased [26,27,28] muscle activation following resistance training. However, cross-sectional studies examining these possible differences including subjects with different resistance training experience have been examined in other resistance exercises [29,30,31]. To the authors’ best knowledge, no previous study has examined barbell free-weight bench press. Therefore, the aim of this study was to examine the effects of grip width on 6-RM loads and muscle activation among a resistance-trained (RT) population and novice-trained (NT) population in bench press using narrow, medium, and wide grip widths. We hypothesized greater triceps brachii, but lower biceps brachii activation using a wide grip compared to a narrow grip width, in addition to lower 6-RM width decreasing grip width for both groups.

## 2. Materials and Methods

### 2.1. Subjects

Twenty-eight men familiar with the barbell free-weight bench press exercise were allocated to the resistance-trained group (RT; n = 15) or the novice-trained group (NT; n = 13). To be included in the study, two inclusion criteria had to be met: (1) The subjects in the RT group had to bench press at least 125% of their body mass [32], whereas the subjects in the NT group had to bench press less than 80% of their body weight. (2) The NT group could not have conducted resistance training regularly in the last six months (i.e., twice or more per week) before being enrolled in the study, whereas the RT group had to have participated in regular resistance training (i.e., twice or more per week) for at least one year before being enrolled. In addition, the subjects had to be free of injuries or pain that could reduce maximal effort. Subjects were excluded from the study if they reported muscular pain, illness, or injury that might affect their maximal effort during testing. The anthropometric descriptions of the subjects are presented in Table 1. 

All subjects were informed both orally and in writing about all testing procedures and potential risks and benefits of participating in the study. Before being enrolled in the study, the subjects had to give their written informed consent. All subjects were over 18 years, and the study conformed with the Norwegian laws, the latest version of the Helsinki declaration and was approved by the Norwegian Centre for Research Data.

### 2.2. Study Design

The study was a within-subjects crossover design comparing the electromyographic (EMG) activity in barbell bench press with narrow, medium, and wide grips. Two groups with different resistance training experiences were recruited. All subjects attended two familiarization sessions and one experimental session. In the first familiarization session the subjects were familiarized with lab settings, warm-up procedures and were tested in one repetition maximum in barbell bench press. In the second familiarization session, the subjects were tested in 6-RM using narrow, medium, and width grips. The order was randomized for each subject and was identical in the familiarization and experimental sessions. In the experimental session, the participants lifted 6-RM in the three grip widths, while the EMG of the pectoralis major (sternocostal and clavicular part), anterior deltoid, medial deltoid, posterior deltoid, triceps brachii (long head), biceps brachii, and latissimus dorsi were obtained. Five days separated each of the three sessions.

### 2.3. Procedures

The subjects were not allowed to train the upper body 48 h before the familiarization- or experimental sessions. The subjects were not allowed to use performance enhancing clothing or equipment (e.g., belt, bench press shirt, or elbow- or wrist straps). The barbell was a standard 20 kg Olympic barbell (Eleiko, Tampere, Finland) with a flat bench (Pivot 430 flexi bench, Sportsmaster, Asker, Norway). Two lab assistants acted as spotters on each side and contributed with the barbell lift-off according to the subjects’ preferences.

Each session had the same warm-up procedures. First, ten minutes of low intensity aerobic exercise (cycling, walking, or jogging) was performed. In the first session, the subjects were asked to estimate or report their 1-RM loads with self-selected grip width. The warm-up consisted of 10 repetitions using 50%, 4 repetitions using 70%, 2 repetitions using 80%, and 1 repetition using 90% of the 1-RM loads. The warm-up sets were separated by two-and-a-half-minute pauses.

After the warm-up in the first familiarization session, 1-RM was tested. The lift started with fully extended elbows. The barbell was lowered to lightly touch the chest (no bouncing was allowed) before being elevated back to the starting position. The head, shoulders, and buttocks had to be in contact with the bench during the entire lift. The width of the feet was self-selected but was measured and controlled before each attempt. The loads were increased by 2.5–5.0 kg until failure or the subject and test leaders agreed that they could not lift more [25]. Four to five minutes separated each 1-RM lift and no more than three attempts were needed [2].

In the second familiarization session, 6-RM with narrow, medium, and wide grip widths were tested. The order of the grip widths was randomized and counterbalanced. When changing grip width, three non-fatiguing repetitions with approximately 50% of self-reported 6-RM loads were lifted with the respectively grip width. The procedures of the grip widths were as the one described in Saeterbakken et al. [4]. Briefly, the subjects stood backs against a wall with the arms elevated in horizontal position with the palms facing towards the floor. A pen was used to mark the point of intersection between the latissimus dorsi and triceps brachii of each side. The distance between the arm pits was measured and defined as narrow grip (i.e., distance between index fingers). Wide grip was defined as the widest grip allowed in bench press competition (i.e., 81 cm). Medium grip width was defined as half the distance between narrow and wide grip. After the individually adjusted grip widths, the 6-RM in each of the three grip widths was achieved within maximal three attempts. The first repetition started by lowering the barbell and lightly touch the chest (no bouncing was allowed) before elevating it to the starting position with fully extended elbows. The same procedures were repeated for each of the repetitions. With the exception of the grip widths, the procedures were identical. Five minutes separated each attempt and each grip width.

### 2.4. Experimental Session

In the experimental session, the 6-RM loads of the three grip widths were tested with EMG activity. The 6-RM loads in the second familiarization session were used to calculate the warm-up loads in the experimental session. After the warm-up, the 6-RM loads lifted in familiarization were added as weight. Loads were decreased or increased by 2.5–5.0 kg if the subjects failed to perform six repetitions or believed they could lift more. Similar procedures were used in the three grip widths.

### 2.5. Electromyography

Before attempting the 6-RM lifts, the subjects’ skin was shaved, abraded, and washed with alcohol before placing the electrodes. Bipolar silver chloride surface electrodes (AE-131 NeuroDyne Medical, Cambridge, MA, USA) with 11-mm contact diameter and 2.0 cm center-to-center distance were placed over the center of the muscle belly along the principal direction of the muscle fibers of the pectoralis major (sternocostal and clavicular part), anterior deltoid, medial deltoid, posterior deltoid, triceps brachii (long head), biceps brachii, and latissimus dorsi. The procedures and placement were performed in accordance with the recommendations of SENIAM [33] and previous studies [4,15,34,35]. The electrodes for pectoralis major were placed 4 cm medial to the axillary fold with the sternocostal part approximately 4 cm above the clavicular part. The electrode for the anterior deltoid was placed one finger width distal and anterior to the acromion. For the medial deltoid, the electrode was placed along the line from the acromion to the lateral epicondyle of the elbow, whereas for the posterior deltoid the electrode was placed approximately two fingerbreadths behind the angle of the acromion. The electrode for the triceps brachii long head was placed at 50% of the line between the posterior crista of the acromion and the olecranon at two finger widths medial to the line. For biceps brachii, the electrode was placed on the line between the medial acromion and the fossa cubit at 1/3 from the fossa cubit. The electrode from latissimus dorsi was placed approximately 4 cm below the interior border of scapula, half the distance between the spine and the lateral edge of the torso. All electrodes were placed on the dominate side (i.e., the preferred throwing arm) of the body [36].

A commercial EMG recording system (Musclelab 4020e, Ergotest Technology A/S, Langesund, Norway) was used. The signals were amplified and filtered using a preamplifier (common rejection rate of 100 dB) located as near the pick-up point as possible. The signals were band pass filtered using a fourth-order Butterworth filter with a high cut-off frequency of 600 Hz and low cut-off frequency of 8 Hz. The EMG signals were converted to root-mean-square using a hardware circuit network (frequency response 0–600 Hz, averaging constant 100 ms, total error ± 0.5%). The RMS converted signal was sampled at 100 Hz using a 16-bit A/D converter.

To identify the start and stop of each repetition, a linear encoder (Ergotest Technology A/S, Langesund, Norway) was used. The linear encoder was placed underneath the barbell and measured the vertical displacement and the time. The encoder had a sampling frequency of 100 Hz and was synchronized with the EMG recordings system using the Musclelab 4020e synchronize unit. The commercial software MuscleLab V8.13 (Ergotest Technology A/S, Langesund, Norway) was used to analyze the EMG recordings and the total time lifting during 6-RM with each grip width. The EMG activity was calculated as the mean of each of the six repetitions, whereas small pauses between repetitions were removed [11,25]. Total time in each grip conditions was calculated by summing up the time for each repetition. The EMG signals from each muscle were normalized using EMG recordings from the subjects three seconds maximal isometric voluntary contraction (MVIC) of each muscle. Bench press (90-degree elbow angle) (pectoralis major) flies (anterior deltoid), rear flies (posterior deltoid), pull-down (latissimus dorsi), push-down (90-degree elbow angle) (triceps brachii), arm curl (90-degree elbow angle) (biceps brachii), and side delt (45-degree elevated from the body) (medial deltoid) were used as exercises [1,37].

## 3. Statistical Analysis

A repeated-measures analysis of variance (ANOVA) with Bonferroni post hoc corrections were used to assess differences in 6-RM, lifting time, and EMG activity between the three grip widths (narrow, medium, and wide) for all the subjects and for the two sub-groups (RT group and NT group). In addition, the intraclass coefficient was calculated between the MVIC attempts for each muscle. All calculations were performed using SPSS (version 27.0; SPSS Inc., Chicago, CA, USA). The statistical significance level was set at *p* < 0.05. The results are presented as mean (±95% confidence intervals) and with Cohen’s d effect size (ES). An ES of <0.2 was consider trivial, 0.2–0.5 as small, 0.5–0.8 as medium, and >0.8 as large [38].

## 4. Results

### 4.1. Subgroup Analyses of 6-RM Loads and Lifting Time

For the 6-RM loads, differences were observed for the RT group (F = 32.170, *p* < 0.001) and NT group (F = 6.027, *p* = 0.010). Post hoc analysis displayed 7.0% and 7.1% lower 6-RM loads with a narrow grip compared to both medium (*p* < 0.001, ES = 0.55) and wide grips (*p* < 0.001, ES = 0.55), whereas no differences were observed between medium and wide grips (*p* = 1.000) for the RT group (Figure 1a). For the NT group, 8.4% and 6.6% lower 6-RM loads were observed using the narrow grip compared to medium (*p* = 0.040, ES = 0.41) and wide grips (*p* = 0.045, ES = 0.35). No difference was observed between medium and wide grips (*p* = 1.000; Figure 1b).

For the total lifting total time, no differences were observed between grip widths for the RT group (F = 3.388, *p* = 0.051) or NT group (F = 1.205, *p* = 0.311).

### 4.2. Subgroup Analyses of Electromyographic Activity

For the RT group, differences between grip widths were observed in triceps brachii (F = 6.178, *p* = 0.012) and biceps brachii (F = 33.541, *p* < 0.001), whereas no differences were observed in the other muscles (F = 0.007–1.649, *p* = 0.220–0.985). For the NT group, differences between grip widths were observed in the anterior deltoid (F = 5.011, *p* = 0.045) and biceps brachii (F = 6.281, *p* = 0.012), whereas no differences were observed in the other muscles (F = 0.301–2.351, *p* = 0.125–0.743).

In triceps brachii, similar muscle activation was observed in the NT group comparing the three grip widths (*p* = 0.189–1.000, Figure 2, Table 2). For the RT group, similar triceps activation was observed between narrow and medium grips (*p* = 1.000). Furthermore, wide grip width led to 10.6% lower activation than a medium grip (*p* = 0.032, ES = 1.48) and 24.1% non-significantly lower than narrow grip, (*p* = 0.058, ES = 0.77, Figure 2, Table 2).

For the anterior deltoid, similar muscle activation between the grip widths were observed in the RT group (*p* = 0.532–1.000). In the NT group, similar EMG activity was observed between the medium and wide grips (*p* = 1.000) and between the narrow and wide grips (*p* = 0.135). In addition, a narrow grip showed 11.7% lower muscle activation than a medium grip (*p* = 0.042, ES = 1.53, Figure 2, Table 2).

In biceps brachii, lower muscle activation was observed with more narrow grip widths for the RT group. Wide grip showed 115.6% (*p* < 0.001, ES = 3.03) and 44.9% (*p* < 0.001, ES = 1.55) greater muscle activation than narrow and medium grip (Figure 2). In addition, medium grip showed 48.8% greater muscle activation than narrow grip (*p* = 0.005, ES = 1.36, Table 2). For the NT group, similar muscle activation was observed between narrow and medium grip widths (*p* = 0.141) and between medium and wide grip widths (*p* = 1.000). Comparing narrow and wide grip widths, a wide grip showed 66.5% greater muscle activation than a narrow grip (*p* = 0.018, ES = 2.36, Figure 2, Table 2).

Comparing the muscle activation in the pectoral major (clavicula part; Figure 2) and sternum; part Figure 2), deltoid posterior, deltoid medial, and latissimus dorsi, between the three grip widths, no differences were observed in the RT group (*p* = 0.069–1.000; Table 2) or the NT group (*p* = 0.141–1.000; Table 2).

The intraclass coefficient values between the two MVCs test for the different muscles were 0.962–0.983 (RT group) and 0.925–0.994 (NT group).

### 4.3. Merged Groups Analysis of Electromyographic Activity

In biceps brachii, triceps brachii and anterior deltoid, differences were observed between the grip widths (F = 11.040–40.085, *p* < 0.001–0.003). Post hoc analyses displayed increasing muscle activity with increasing grip width (*p* < 0.001–0.001, ES = 0.38–0.69, Table 2). Lower triceps brachii activity was observed in the wide grip compared to medium (*p* = 0.004, ES = 0.27) and narrow grip widths (*p* = 0.008, ES = 0.32), whereas no difference was observed between narrow and medium grips (*p* = 0.897). In the anterior deltoid, greater muscle activation was observed using medium than narrow grip (*p* = 0.036, ES = 0.14). No difference was observed between the narrow and wide grip (*p* = 0.328) or between medium and wide grips (*p* = 0.996). For the other muscles, no differences were observed between the grip widths (F = 0.660–2.628, *p* = 0.117–0.424, Table 2).

## 5. Discussion

Merging the two groups, the three grip widths showed differences in biceps brachii (increased with increasing grip widths) and lower triceps brachii muscle activation using a wide grip compared to a narrow grip width. For the other muscles, similar muscle activation was observed. For the sub-group analyses, narrow grip width reduced 6-RM loads compared to medium and wide grip width for both groups, whereas no differences were observed between the medium and wide grip widths. In general, the RT group demonstrated similar differences across the grip widths as the main analyses, whereas the NT group only displayed greater biceps brachii activity in wide vs. narrow grips and lower anterior deltoid activity in narrow compared to medium grip widths.

Similar muscle activity was observed in pectoralis major, latissimus dorsi, medial, and posterior deltoid between the three grip widths independent of resistance training experience or merging the groups. Similar muscle activation between the grip widths may be a result of similar relative intensity tested (i.e., 6-RM), despite differences in absolute loads. Furthermore, the difference between the grip widths might have been too small to affect the position of the glenohumeral joint position. For example, previous studies have demonstrated greater pectoralis major activity using a grip width of 190–200% of the biacromial distance compared to 100% of the biacromial distance [12,16,17]. In the present study, a wide grip was defined as the greatest grip width allowed in bench press competition (i.e., 81 cm) to increase the ecological validity. Of note, none of the comparable studies [12,16,17] reported the subjects’ biacromial distance. Therefore, we can only speculate that the present study’s wide grip was narrower than in previous studies [12,16,17]. Furthermore, comparable muscle activity was observed among competitive bench press athletes using grip widths similar to the present study [4]. In contrast, Barnett and colleagues [17] and Mausehund et al. [9] demonstrated greater pectoralis major activity (clavicular part) using a narrow grip compared to a wide grip.

In agreement with the hypotheses, lower triceps brachii activity was observed using a wide grip compared to the other grip widths with greater biceps brachii activation with increasing grip widths when the groups were merged. Similar findings were also observed in the RT group, whereas the NT group did not demonstrate differences in triceps brachii and only displayed differences in biceps brachii between narrow and medium grip widths. However, and according to the recommendations from Vigotsky et al. [39], statistical analyses comparing the muscle activity between the groups were not conducted. Visually comparing the normalized muscle activity independent of grip width (Table 2), the results support previous studies reporting increased neural drive as neurological adaptions to resistance training [26,27,28]. The differences in biceps and triceps brachii activation in the NT group between the three grip widths were not clear. One might speculate that the training experience might explain the differences and that a certain level training experience, skills, and technique is required in bench press and muscle activations [22]. Using a narrow grip, the elbows adduct towards the body, causing a reduced transverse flexion of the shoulder joint. As a result of a narrower grip, one could expect a lower elbow angle which might be the cause of the reduced biceps activation [9,40]. Furthermore, decreased biceps brachii activation using a narrow grip could reflect the inhibitory drive of the triceps brachii to limit a co-contraction. Inhibitory muscle activity in the antagonist has been displayed as training related adaptions [22] and might explain the findings in the RT group. In the NT group, similar triceps brachii activation was displayed between the grip widths and only greater activation of the biceps brachii was observed using wide grip compared to narrow grip. However, Mausehund et al. [9] displayed larger elbow joint momentum and larger triceps brachii activation with decreasing grip width in bench press. Of note, the authors included two narrow grip conditions (elbows close to or away the body) which limited the generalizing from previous and present findings [9]. It is therefore possible that the elbow momentum might influence elbow flexors and extensors, and future studies should examine this.

In general, the mean normalized muscle activity in the prime movers (i.e., triceps brachii, deltoid anterior, and pectoralis major) was 23–80% greater between grip widths in the RT group than in the NT group. In other words, the differences observed between groups were not dependent on grip widths. Whether the visually observed differences between the two groups were caused by increased firing frequency, motor unit activation, or synchronization remains purely speculations. Importantly, both groups displayed excellent reliability [41] in the MVCs of the different muscles with intraclass coefficient values between 0.962–0.983 (RT group) and 0.925–0.994 (NT group). Lack of synchronization between posterior or anterior deltoid may explain greater anterior deltoid activation using a medium grip compared to narrow in the main analyses and NT group. Alternatively, the subjects did manage to maintain their shoulder arching position using the narrow grip which may cause greater shoulder adduction and thereby greater muscle activation. Finally, the techniques or motor behavior may explain the findings. Using the narrow grip, the NT group may lower the barbell by using elbow flexion to a greater extent more than extending the shoulder joint. Furthermore, the RT group displayed similar anterior deltoid activation across the grip width which has been reported previously [4]. In contrast, Barnett et al. [17] and Mausehund et al. [9] reported lower anterior deltoid activation using a wide grip compared to a narrow grip. However, the cause of these differences was uncertain but could be related to the shoulder and elbow momentum [9].

As hypothesized, greater 6-RM loads were observed with wider grip width in both groups even though none of the groups displayed differences between medium and wide grip width. The reduced vertical displacement of the barbell with a wider grip is the most obvious explanation and similar findings have been observed among competitive bench press athletes and resistance-trained subjects [4,8]. The findings were in contrast to Barnett and colleagues [17], who demonstrated a 5% non-significant reduction in loads lifted between narrow and wide grip widths. Importantly, Barnett et al. [17] only recruited six participants and low statistical power may explain the lack of significant difference between grip widths. In the present study, wide grip was most used by the participants (especially the RT-group) in their normal training routines. Therefore, the task-specificity of the preferred and/or most familiar grip width compared to the other grip widths may be used as a secondary explanation [20,21,25,42]. Based on the two groups’ training experience, one would have expected greater reduction in 6-RM strength with narrow grip in the RT group than the NT group. Both groups demonstrated approximately the same percentage reduction (6.6–8.4%) between grip widths. This leads to the speculation that narrow grips causes poor mechanical force positions that inhibit the capacity to exert torque.

The present study has some limitations that need to be addressed. For example, measurement of morphological characteristics (e.g., cross-section area, muscle fiber length, fiber pennation angle) were not included. Both the neurological and morphological adaptions to resistance training needs to be considered when comparing different resistance training experiences. Importantly, identical testing procedures were conducted between the groups. Furthermore, being novice and thereby not familiarized with testing and/or training to fatigue (e.g., 6-RM loads), it is possible that the NT group should have attended more than only one familiarization session using 6-RM loads. Unfortunately, the 1-RM and 6-RM results from the familiarization sessions were lost and we could therefore not calculate the intraclass coefficients between sessions. Finally, the aim of the study was to examine novice-trained compared to resistance-trained subjects, and therefore we acknowledge that the technique, performance, and familiarization to bench press procedures would differ from the resistance training group.

## 6. Practical Implication

Studies have indicated that a narrow grip may be better suited for specific movements in sports [43,44]. In addition, a closer grip (i.e., <1.5 biacromial width) produces less stress for the acromioclavicular joint, decreases the angles of shoulder abduction, and thereby reduces the risk of injuries [45,46]. The present study displayed decreased 6-RM loads with narrow grip compared to the wider grips for both the resistance trained group and the NT group. This may decrease the stress of the shoulder joint over time. However, using a wider grip increased the 6-RM and thereby the mechanical loading which is an important stimulus for increasing muscle strength and hypertrophy. Furthermore, only minor differences were observed in muscle activity pattern between the grip widths independent of training status. With the exception of biceps brachii, all grip widths can be included in training programs aiming to optimize muscle activity. Visually comparing the normalized muscle activity between the two groups, greater muscle activity was observable in almost every muscle independent of grip width. This may indicate that resistance-trained subjects are better at recruiting their motoneurons compared to novice lifters.

## 7. Conclusions

In general, the main analyses in narrow, medium, and wide grip widths in bench press displayed greater biceps brachii, but lower triceps brachii activation with increasing grip widths, whereas the remaining muscles displayed similar muscle activity across grip widths with the exception of deltoid anterior. The resistance training experience demonstrated a similar muscle activation pattern across the grip widths with some deviation between groups in biceps brachii, triceps brachii, and deltoid anterior. Among resistance-trained and novice-trained subjects, 6-RM loads were heavier with medium and wide grips compared to a narrow grip, but there were no differences between the medium and wide grip width. Longitudinal variation in grip widths may stimulate hypertrophy or strength gains in the elbow flexors-and extensor differently whereas the grip widths may induce similar adaptions for the other muscles. In general, conclusions based on training effects with EMG measurements should be interpreted with cautions.

## Figures and Tables

**Figure 1 ijerph-18-06444-f001:**
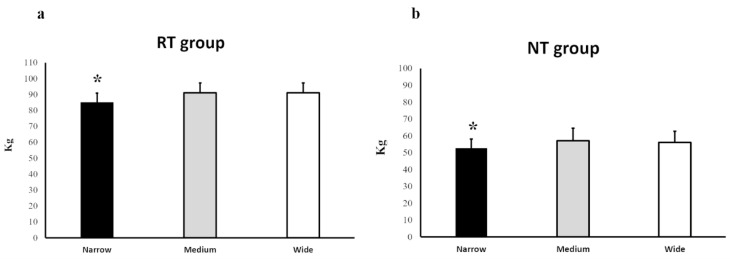
(**a**,**b**). The mean 6-RM loads (95% CI) in narrow, medium, and wide grip widths for the RT group (**a**) and novice group (**b**). * Significant difference compared to the other grip widths.

**Figure 2 ijerph-18-06444-f002:**
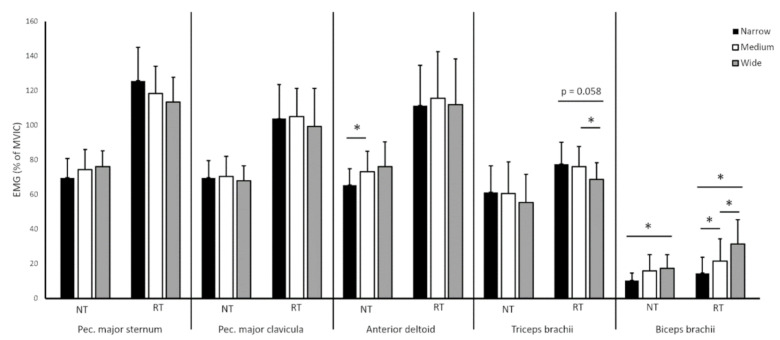
The normalized EMG activity (i.e., % of MVIC) for the RT group and NT group as mean (95% CI). * Significant difference between the grip widths.

**Table 1 ijerph-18-06444-t001:** The anthropometric of the subjects (mean ± standard deviation).

Group	Age (Years)	Height (cm)	Weight (kg)	Resistance Training Experience (Years)	Relative Strength (6-RM Wide Grip/Body Weight)
RT	23.7 ± 2.0	180 ± 4.8	81.7 ± 8.2	5.5 ± 1.9	1.11
NT	23.7 ± 3.8	183 ± 6.2	79.0 ± 9.3	1.0 ± 0.7	0.79

RT = resistance trained, NT = novice trained, cm = centimeters, kg = kilograms.

**Table 2 ijerph-18-06444-t002:** Normalized EMG activity (i.e., % of MVIC) from the three grip widths narrow, medium, and wide (mean ± 95% CI).

Grip Width	Group	Latissimus Dorsi	Biceps Brachii	Triceps Brachii	Posterior Deltoid	Medial Deltoid	Anterior Deltoid	Pectoralis Major Clavicula	Pectoralis Major Sternum
**Narrow**	**All**	21.4 (15.2–27.5)	12.6 (7.6–17.7) *	70.3 (60.7–79.9)	11.6 (7.3–15.9)	29.4 (23.8–35.0)	90.0 (74.7–105.4) ‡	88.0 (75.2–100.7)	100 (85.3–115.2)
	NT	16.3 (10.8–21.8)	10.4 (6.2–14.6) #	61.3 (46.0–76.6)	8.3 (5.9–10.7)	30.0 (22.1–38.0)	65.4 (55.8–75.0) ‡	69.7 (59.7–79.7)	69.7 (58.7–80.8)
	RT	25.8 (15.2–36.4)	14.6 (5.3–23.8) *	77.6 (65.0–90.1) *	14.40 (6.5–22.3)	28.9 (20.2–37.6)	111.4 (88.1–134.6)	103.8 (84.0–123.6)	125.7 (106.5–145.0)
**Medium**	**All**	23.3 (17.0–29.7)	19.0 (11.3–26.7) #	69.2 (59.1–79.3)	9.9 (7.8–11.9)	30.8 (23.8–37.8)	95.8 (79.1–112.6)	89.0 (77.2–95.1)	97.9 (85.2–110.7)
	NT	18.3 (12.0–24.6)	15.9 (6.6–25.3)	60.5 (42.1–78.9)	7.6 (5.3–9.8)	32.8 (19.5–46.2)	73.1 (61.0–85.1)	70.5 (59.0–82.0)	74.5 (63.1–85.9)
	RT	27.7 (17.0–38.5)	21.7 (8.8–34.5)	76.1 (64.6–87.6)	11.9 (8.7–15.0)	29.0 (21.0–37.0)	115.53 (88.7–142.4)	105.0 (88.7–121.3)	118.3 (102.4–134.1)
**Wide**	**All**	24.1 (17.2–31.0)	24.9 (16.5–33.2)	62.7 (54.2–71.6) *	9.6 (7.8–11.4)	31.2 (24.9–37.6)	95.3 (79.0–111.5)	84.8 (74.4–95.1)	96.1 (85.1–107.1)
	NT	17.6 (12.0–23.7)	17.3 (9.3–25.3)	55.5 (39.3–71.7)	7.7 (5.3–10.1)	33.6 (21.9–45.3)	76.1 (61.7–90.5)	68.0 (59.2–76.7)	76.1 (66.8–85.3)
	RT	29.7 (17.9–41.5)	31.4 (17.3–45.5) ‡	68.8 (59.2–78.4) ‡	11.3 (8.7–13.9)	29.1 (21.6–36.7)	111.9 (85.5–138.2)	99.3 (88.7–121.3)	113.5 (99.4–127.6)

RT = resistance trained. NT = novice trained. * Significant difference from the other the grip widths. # Significant difference from the wide grip width. ‡ Significant difference from the medium grip width.

## Data Availability

The raw data supporting the conclusions will be made available by the authors without undue reservation.

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
