# Peer review of "The Effect of Grip Width on Muscle Strength and Electromyographic Activity in Bench Press among Novice- and Resistance-Trained Men"

_ijerph, 2021, doi:10.3390/ijerph18126444_

Round 1

Reviewer 1 Report

General Comments

The paper addresses a useful question on the effect of different grip widths on muscle strength and EMG activity in bench press exercise also making a comparison between novice and experienced resistance trained men. The information provided in the Introduction section was overall clear and justified the methodological as well as the innovative aspects of the study. The authors investigated the changes in mean EMG activity among three different grip widths (i.e. narrow, medium, and wide) normalizing the EMG signal by MVIC for each of the investigated muscles. Sufficient familiarization for proper task execution was provided, especially considering the involvement of novice-level participants. Methodological choices were justified throughout the Methods section and proper considerations were also made in the Discussion, which makes the methodology basis of the present work sound. Results were clearly reported, along with statistical results, and discussed within the framework of the bench press performance and biomechanics obtained from the most updated literature. Overall, the manuscript is well-written and only minor changes are needed in order to consider the paper for publication.

Introduction

Lines 33-37: these sentences about the effect of training experience are placed between the introduction and description of the bench press exercise. The authors should consider to move them later in the introduction (i.e. last paragraph), when differences between novices and experienced athletes are presented.

Line 50: please remove “the”.

Lines 75-76: the hypothesis of different level of EMG activity between experienced and novice groups is made, but there is no reference about this analysis in the Methods section. In the Discussion section, the authors declared that this analysis was not performed according to the recommendations from Vigotsky. Therefore, for the sake of clarity, I would suggest to remove the hypothesis from the Introduction, still discussing this point in the Discussion section.

Results

ICC analysis is mentioned in both Methods and Discussion sections but results are reported directly while discussing the results. Please, report ICC analysis results also in the Results section.

Minor issues

Abundant use of “However” and “Still” is made throughout the manuscript even when not really appropriate. Please, amend this.

Author Response

Thank you for reviewing the manuscript and for the many constructive comments and suggestions for improvement. We hope that the changes we have made have improved the manuscript.

The paper addresses a useful question on the effect of different grip widths on muscle strength and EMG activity in bench press exercise also making a comparison between novice and experienced resistance trained men. The information provided in the Introduction section was overall clear and justified the methodological as well as the innovative aspects of the study. The authors investigated the changes in mean EMG activity among three different grip widths (i.e. narrow, medium, and wide) normalizing the EMG signal by MVIC for each of the investigated muscles. Sufficient familiarization for proper task execution was provided, especially considering the involvement of novice-level participants. Methodological choices were justified throughout the Methods section and proper considerations were also made in the Discussion, which makes the methodology basis of the present work sound. Results were clearly reported, along with statistical results, and discussed within the framework of the bench press performance and biomechanics obtained from the most updated literature. Overall, the manuscript is well-written and only minor changes are needed in order to consider the paper for publication.

 Thank you.

Introduction

Lines 33-37: these sentences about the effect of training experience are placed between the introduction and description of the bench press exercise. The authors should consider to move them later in the introduction (i.e. last paragraph), when differences between novices and experienced athletes are presented.

It’s a good point. We have moved the sentences to last paragraph as suggested.

Line 50: please remove “the”.

“The” has been deleted. Thank you and we apologize for this typing error.

Lines 75-76: the hypothesis of different level of EMG activity between experienced and novice groups is made, but there is no reference about this analysis in the Methods section. In the Discussion section, the authors declared that this analysis was not performed according to the recommendations from Vigotsky. Therefore, for the sake of clarity, I would suggest to remove the hypothesis from the Introduction, still discussing this point in the Discussion section.

We agree, and have removed the hypothesis from the introduction, but still discussing this point in the discussion section.

Results

ICC analysis is mentioned in both Methods and Discussion sections but results are reported directly while discussing the results. Please, report ICC analysis results also in the Results section.

We apologize for this mistake. The ICC values have been included in the results as suggested.

Minor issues

Abundant use of “However” and “Still” is made throughout the manuscript even when not really appropriate. Please, amend this.

Thank you for brining this to our attention. We have made changes throughout the manuscript.

Reviewer 2 Report

This study compared the muscle activity and six repetition maximum (6- 11 RM) loads in bench press with narrow, medium, and wide grip widths with sub-group comparisons of resistance trained (RT) and novice men.

The study has been well conducted, in general, although a series of aspects are suggested for a better understanding of the reviewer and the readers.

1. A complete revision of the text is recommended to correct typographical errors (eg, end of line 83, 188 ..).

2. It is suggested to also use an acronym for "novice group".

3. It is suggested, in the abstract, to name the muscles that have been analyzed electromyographically.

4. A concrete explanation is suggested of how the participants of the novice group were distributed, that is, by lifting less than 80% of their body weight or by years of experience / "different resistance training experiences"?).

5. It is suggested to include, in Table 1, information on the weight lifted in the 6 RM, with each of the different grips.

6. It is necessary to explain if in the experimental session the movement was also eccentric-concentric.

7. It is suggested, in section 2, a subsection dedicated to the analysis of the lifting time, since it is an analysis that is presented in the Results section.

8. The authors should clarify with greater precision, for better understanding of the readers, when the electromyographic analysis was performed between the three types of grips, when it was performed between the two groups, or when it was performed combined between the three types of grips and the two groups; and, if one of these analyzes was not carried out, clarify it in the text (sometimes it is not possible to differentiate what type of analysis is being carried out, even in the titles of the subsections).

9. On the contrary, Table 2 explains very clearly what has been analyzed.

10. The same happens in the Discussion: for example, it is stated that "The three grip widths showed differences in biceps brachii (increased with increasing 275 grip widths) and lower triceps brachii muscle activation using wide grip compared to narrow grip width". If this occurs for both groups, it must be specified. I recommend to the authors a global reading, and clarify when the analysis refers to the three grips in a single group, or to a specific grip in both groups, or to a muscle in all three grips, or to a muscle, a grip and both groups. It is really sometimes very confusing to know what the authors are referring to, because three grips, two groups and several muscles are combined in the analysis.

11. Although the authors refer to certain limitations of the study (with which I agree), I recommend not mixing them with the discussion, but in a separate paragraph.

Author Response

Thank you for reviewing the manuscript and for the many constructive comments and suggestions for improvement. We hope that the changes we have made have improved the manuscript.

This study compared the muscle activity and six repetition maximum (6- 11 RM) loads in bench press with narrow, medium, and wide grip widths with sub-group comparisons of resistance trained (RT) and novice men.

The study has been well conducted, in general, although a series of aspects are suggested for a better understanding of the reviewer and the readers.

  1. A complete revision of the text is recommended to correct typographical errors (eg, end of line 83, 188 ..).

Thank you for bringing this to our attention. We have carefully revised the manuscript for typographical errors, and we apologize for these mistakes. Minor typographical changes have not been highlighted in the manuscript.

  1. It is suggested to also use an acronym for "novice group".

Novice group has been named NT (Novice trained) in the manuscript, as suggested.

  1. It is suggested, in the abstract, to name the muscles that have been analyzed electromyographically.

We agree, but due to the numbers of words allowed in the abstract (maximal 200 words according to the instructions for authors), we were not able to include them.

  1. A concrete explanation is suggested of how the participants of the novice group were distributed, that is, by lifting less than 80% of their body weight or by years of experience / "different resistance training experiences"?).

Thank you for brining this to our attentions. We understand the lack of clarity and have therefore re-written this section.

  1. It is suggested to include, in Table 1, information on the weight lifted in the 6 RM, with each of the different grips.

All the 6-RM loads are presented in Figure 1 in the result section. However, we agree to your suggestion to add this information twice. To increase the insight in the participants strength beyond what`s presented in figure 1, the relative strength (6-RM divided by the body weight) has been added to in the Table 1.

  1. It is necessary to explain if in the experimental session the movement was also eccentric-concentric.

Of course, the information is added in the manuscript (paragraph 2.3 procedures).

  1. It is suggested, in section 2, a subsection dedicated to the analysis of the lifting time, since it is an analysis that is presented in the Results section.

More details of the lifting time have been included in paragraph 2.5, as suggested.

  1. The authors should clarify with greater precision, for better understanding of the readers, when the electromyographic analysis was performed between the three types of grips, when it was performed between the two groups, or when it was performed combined between the three types of grips and the two groups; and, if one of these analyzes was not carried out, clarify it in the text (sometimes it is not possible to differentiate what type of analysis is being carried out, even in the titles of the subsections).

We have tried to provide a better clarity of the comparisons between groups and grip withs in the manuscript to improve the readability.

  1. On the contrary, Table 2 explains very clearly what has been analyzed.

Thank you.

  1. The same happens in the Discussion: for example, it is stated that "The three grip widths showed differences in biceps brachii (increased with increasing 275 grip widths) and lower triceps brachii muscle activation using wide grip compared to narrow grip width". If this occurs for both groups, it must be specified. I recommend to the authors a global reading, and clarify when the analysis refers to the three grips in a single group, or to a specific grip in both groups, or to a muscle in all three grips, or to a muscle, a grip and both groups. It is really sometimes very confusing to know what the authors are referring to, because three grips, two groups and several muscles are combined in the analysis.

Please see the discussion. We have tried to re-written parts to improve the understanding and clarify the analyses being discussed.

  1. Although the authors refer to certain limitations of the study (with which I agree), I recommend not mixing them with the discussion, but in a separate paragraph.

A separated paragraph has been included.